# Utilizing Fish Skin of Ikan Belida (*Notopterus lopis*) as a Source of Collagen: Production and Rheology Properties

**DOI:** 10.3390/md20080525

**Published:** 2022-08-18

**Authors:** Tzen T. Heng, Jing Y. Tey, Kean S. Soon, Kwan K. Woo

**Affiliations:** 1Department of Chemical Engineering, Lee Kong Chian Faculty of Engineering and Science, Universiti Tunku Abdul Rahman, Jalan Sungai Long, Bandar Sungai Long, Cheras, Kajang 43000, Selangor, Malaysia; 2Department of Mechanical Engineering, Lee Kong Chian Faculty of Engineering and Science, Universiti Tunku Abdul Rahman, Jalan Sungai Long, Bandar Sungai Long, Cheras, Kajang 43000, Selangor, Malaysia

**Keywords:** collagen, alginate, hydrogels, biomaterials

## Abstract

Collagen hydrogels have been extensively applied in biomedical applications. However, their mechanical properties are insufficient for such applications. Our previous study showed improved mechanical properties when collagen was blended with alginate. The current study aims to analyze the physico-chemical properties of collagen-alginate (CA) films such as swelling, porosity, denaturation temperature (*T_d_*), and rheology properties. Collagen was prepared from discarded fish skin of Ikan Belida (*Notopterus lopis*) that was derived from fish ball manufacturing industries and cross-linked with alginate from brown seaweed (*Sargasum polycystum*) of a local species as a means to benefit the downstream production of marine industries. CA hydrogels were fabricated with ratios (*v*/*v*) of 1:1, 1:4, 3:7, 4:1, and 7:3 respectively. FTIR spectrums of CA film showed an Amide I shift of 1636.12 cm^−1^ to 1634.64 cm^−1^, indicating collagen-alginate interactions. SEM images of CA films show a porous structure that varied from pure collagen. DSC analysis shows *T_d_* was improved from 61.26 °C (collagen) to 83.11 °C (CA 3:7). CA 4:1 swelled nearly 800% after 48 h, correlated with the of hydrogels porosity. Most CA demonstrated visco-elastic solid characteristics with greater storage modulus (G′) than lost modulus (G″). Shear thinning and non-Newtonian behavior was observed in CA with 0.4% to 1.0% (*w*/*v*) CaCl_2_. CA hydrogels that were derived from discarded materials shows promising potential to serve as a wound dressing or ink for bio printing in the future.

## 1. Introduction

Collagen is a structural protein that is distributed in the skin, tendons, bones, cartilage, and connective tissues of animals. The collagen structure is built out of a unique triple helix conformation with repeating amino acids sequence of -Gly-X-Y- that are linked to a substantial amount of hydroxyproline and hydroxylysine groups in which the X and Y are often denoted as proline and hydroxyproline. The diversity in collagen characteristics is governed by the hydroxyproline distribution in the collagen molecules. Therefore, the properties of collagen, including thermal stability, rheological behavior, and solubility, varies among animal species and their habitats [1,2,3]. Several types of collagens have been isolated including fibril protein type I, II, III, V, and IX that are used in fabricating biomaterials. Currently, collagens have been extensively isolated from land-based animals such as cows (bovine) and pigs (porcine). These sources are however not as “clean” or safe compared to their marine collagen counterparts. This is due to the possibility of disease transmission such as BSE (bovine spongiform encephalopathy) or mad cow disease [4], and the religious connotations that prevail within the Islamic, Jewish, and Hindu worlds. As such, marine-derived collagens are widely extracted to replace the collagen that is derived from mammals. These collagens may be extracted from the marine by-products such as fish skin, bones, and scales. These sources enable the production of proteins that are often similar to Type I collagen in its properties. Marine sources from fishes have gradually gained attention for biomaterial fabrication collagen to be extracted from fish skin as it is preferable due to its relatively high purity (around 70%) [5].

Type I collagen is the most common type of biomaterial that is applied in tissue engineering for fabricating artificial organ and human mimicry systems. This is because the collagen has low antigenicity, high biodegradability, and biocompatibility [6,7,8,9]. The collagen can also provide structural support to tissues as well as cell recognition sites for cell attachment and migrations [10]. The major advantages of collagen being favorable in biomedical application are due to its ability to self-assemble into fibrils structure in physiological conditions [11,12,13]. Various collagen-based extracellular matrixes (ECM) have been developed using freeze-drying [14], injection [15], or cell encapsulation [16] methods. Currently, the development of a 3D printing method has brought advancement in fabricating complex and personalized tissues and organs. This building block materials for bio fabrication are known as biomaterial inks which can be made into hydrogels from either natural or synthetic polymers. Therefore, developing hydrogels which are sufficient for bio printing purposes is the future aim of our current study.

Unfortunately, the mechanical properties of collagen-based hydrogels are incomparable to synthetic polymers. These deficiencies remain as major challenges in utilizing collagen-based hydrogels in 3D bio printing research. Due to that, collagen hydrogels usually undergo structural modification with chemical cross-linking or combining with other bioactive polymers or compounds that helps to overcome the limitations of pure hydrogels [17,18,19,20]. The modification of collagen by cross-linkage with polysaccharides such as alginate, has been suggested. This is due to its unique structural properties that can improve the toughness and stiffness of pure collagen hydrogels [21]. Combining collagen with alginate will further lead to rheological improvements compared to hydrogel with a single biomaterial [22]. The rheological properties are important to determine the possibility of using cross-linked collagen hydrogels in developing a 3D printing biomaterial ink. The cross-linked collagen hydrogels must possess a low enough viscosity (300 cps to 3000 cps) to ease the extrusion through the printing nozzle during 3D printing. This is important as to maintain its shape after extrusion [23].

Therefore, the characterization of collagen-alginate hybrid hydrogels in a systematic order is essential. The hydrogels must achieve a certain viscosity and mechanical strength upon fabrication to withhold the shape after being extruded through the printing nozzle. Furthermore, seeding different cells in the hydrogels prior to printing will produce a different rheological property. Thus, a hydrogel should be sufficiently stable in structure (prior and post-printing), biodegradable, and nontoxic. Articles that are related to the rheological properties of collagen have been published elsewhere. For example, Zhang et al. [24] fabricated a 3D printed construct using a gelatin-alginate hydrogel from cold water fish skin which showed high cell viability. A study with freshwater fish, Nile tilapia [25], was shown to be successful in the development of wound dressing hydrogels that has improved healing effects on secondary burns. The application of marine collagen in the biomedical field is promising but should be further improved technically and economically.

The current study aims to use fish skin collagen that was extracted from Ikan Belida (*Notopterus lopis*). Ikan Belida is a type of freshwater fish that is found in Malaysia that is famous for making fish cake and fish balls from its meat. Based on the Malaysia Annual Series of Fishery Commodities Export 2018, more than 82 thousand metric tonnes of fish products (including fresh fish, chilled or frozen, fish meat, and fillet) were exported [26]. According to the Fishery Department of Malaysia, approximately 1.38 million tonnes per annum fishes were caught in year 2020 [27]. The volume of fishes that is landed is expected to increase annually. As such, the leftover fish skin, the major by-product from the meat extracting process, from these fish processing industries are expected to be in thousands of metric tonnes annually. An effective waste management mechanism is essential in avoiding environmental pollution. Valorizing fish wastes (fin, bones, scales, and skin) as a downstream production route is indeed crucial in diversifying the revenue of the local fishing industries and reducing the environmental burden.

Our previous study [28] revealed that fish skin contains promising amounts of collagen yield from using an acid extraction method. To further expand the application of fish skin collagen, a hybrid hydrogel of collagen and alginate was developed [29]. The results from the study showed improved mechanical properties of collagen-alginate films [28]. Therefore, the current study aims to further discover the rheological properties of Ikan Belida collagen that was hybridized with brown seaweed (*Sargasum polycystum*) alginate. Alginate is a polysaccharide that is commonly also used as a biomaterial in tissue engineering applications and drug delivery systems. The application of collagen-alginate hydrogels varies among their properties. Collagen-alginate hydrogels have been developed into wound dressing, tissue engineering, tissue regeneration, encapsulated cell therapy, and tumor biology [30]. It has been reported that collagen is able to enhance wound healing due to increased cell proliferation, easily biodegradability, and guiding cell differentiation [31]. The properties of alginate such as low cytotoxicity to mammalian cells and moisture retention capabilities also serve to highlight its potential application as a wound dressing replacement [32].

Alginate (also known as alginic acid) was described as a group of naturally occurring anionic polysaccharides that were obtained from brown algae cell walls [33]. Alginate is made up of two components, β -D- mannuronic acid (M) and α -L-guluronic acid (G). The former (M) determines the viscosity while the latter (G) determines the gelling properties [34]. However, alginate lacks components such as Arginine, Glycine, and Aspartate (RGD) sequences motifs that help in wound healing [35]. This is due to the RGD sites in the collagen that binds integrin of cells, enhance cell adhesion and encourage the movement of fibroblasts and keratinocytes [36].

Our earlier study on fish skin collagen revealed collagen-alginate hybrid hydrogels improved the mechanical properties and denaturation temperature (*T_d_*) [29]. Therefore, the present research attempts to understand the rheological behavior for fish skin collagen, brown seaweed alginate, and collagen-alginate hybrid hydrogels. The effect of calcium chloride on the gelation properties on collagen-alginate hydrogels were also investigated at various concentrations. It is well documented that the viscoelasticity of alginate hydrogels can be regulated via ionic cross-linking with calcium chloride. Furthermore, calcium chloride does not affect or alter the alginate hydrogels biocompatibility and cell viability.

While the improvements that can be gleamed from the mixing of collagen and alginate results in an ideal hydrogel, this positive outlook is expensive to fabricate due to the excessive cost of materials that will be used for such an endeavor. Most studies or research fabricate said hydrogels using pre-bought collagen and alginate powders that are derived from reputable chemical companies that are not easily obtained due to excessive cost. Thus, it is not wrong to search for feasible alternative solutions to this dilemma. In this case, it is cheaper to extract the required biomaterials by themselves

Both extraction methods utilized cheap and common chemicals that can be easily procured for large scale extraction. Hence, this study aims to prove that collagen and alginate that are extracted from fish skin and brown seaweed are viable alternatives. The results may serve as the basis for local fishery industries in developing a downstream processing route and diversifying their income. This is especially important during the monsoon season which fishing activities are almost impossible. This process of reusing discarded waste products from seafood processing also serves an important role in reducing the amount of waste products that are thrown away. The processing of fish produces around 20% to 80% of waste products that are discarded haphazardly [37]. Hence, it is important to educate the general public and fishing companies in the country that there is a potential in extracting value-added product such as collagen from said waste products. This in turn helps to decrease the waste products and develop a secondary economic product for purchase. This, in essence, encourages a circular economy that prevents wastage and produces high value products in a cheaper manner.

## 2. Results

### 2.1. Collagen and Alginate Yield

Collagen was successfully extracted from the skin of Ikan Belida which yielded approximately 0.25% on a dry weight basis using an acidic extraction method. On the other hand, 26.08% sodium alginate yield was obtained from brown seaweed using a cold method.

### 2.2. Fourier Transform Infrared Spectroscopy (FTIR)

Fourier transform infrared (FTIR) analysis was carried out on the collagen, alginate, and collagen-alginate blends. Figure 1 shows the FTIR spectra and their respective characteristic bands and changes of patterns on the collagen-alginate blends as compared to the controls. Fish skin collagen exhibited strong signals at 1636 cm^−1^, 1541.11 cm^−1^, and 1405.56 cm^−1^ corresponding to Amide I, II, and III, respectively. These were the typical bands for protein that was related to stretching vibration of carbonyl CO and CN groups, NH bending vibration, and C-H stretching [38,39]. The vibration band of the N-H group appeared in all the samples containing collagen (Figure 1a–d) across the region of 3408.90 cm^−1^ to 3417.34 cm^−1^. Characteristic bands of alginate were observed in the region of 3295.36 cm^−1^ representing O-H stretching vibration (Figure 1e). A slightly weak signal was observed on 2926.26 cm^−1^ which was attributed to the CH_2_ of alginate, whilst signals in the region of 1596.42 cm^−1^ and 1407.31 cm^−1^ correlated to asymmetric and symmetric stretching modes of -COONa. Vibrations in the region of 1083.63 cm^−1^ represents C-O-C stretching in the glycoside bonds in the polysaccharide [40]. Shift of Amide bands were observed with the addition of alginate to the collagen blend. Amide I changed from 1636.12 cm^−1^ (Figure 1a) to 1634.64 cm^−1^ (Figure 1d), Amide II (from 1541.11 cm^−1^ to 1548.38 cm^−1^), and Amide III (from 1405.56 cm^−1^ to 1408.33 cm^−1^), were observed as the alginate content increases in the gel blend.

### 2.3. Scanning Electron Microscopy (SEM)

The surface morphology of the freeze-dried collagen-alginate film was studied using scanning electron microscope (SEM). Figure 2 shows that the SEM images with the same magnification displays clear morphological changes between the pure collagen, pure alginate, and collagen-alginate hybrids film. Pure collagen appears compact, rigid, dense, and rough with some rod-like structures (Figure 2a). On the other hand, pure alginate film has a porous structure that is arranged in an egg box network conformation with a pore size that ranged from approximately 30 μm to 120 μm in diameter (Figure 2b). Changes of the surface morphology from dense, rigid structure to porous and flaky conformation on the hybrid films (Figure 2c–e) as the alginate concentration gradually increased in the CA films.

### 2.4. Differential Scanning Calorimetry (DSC)

The differential scanning calorimetry (DSC) method was used to determine the thermal stabilities of the CA films (Figure 3). The collagen-alginate films revealed significant (*p* < 0.05) improvement in the denaturation temperature (*T_d_*) from 61.26 °C (collagen) to 83.11 °C (CA 7:3). In contrary, the collagen-alginate blends with higher alginate content (CA 3:7 and CA 1:4) exhibited lower *T_d_*.

### 2.5. Swelling Behavior of Collagen-Alginate Films

Swelling ratios indicated that the water absorption ability of the freeze-dried CA hydrogels. CA 4:1 film showed the highest stability among other CA films. The CA films with higher collagen CA 4:1 maintained until 48 h followed by CA 7:3 and CA 1:1 was able to tolerate soaking up to 24 h (Figure 4a). CA with higher alginate content (CA 3:7) dissolved after 2 h. Pure alginate gels dissolved completely after soaking for less than two hours due to its water solubility properties. This result correlates with the SEM images from Figure 2. The CA 1:1 film shows to best swelling ability compared to the others when it was soaked for extended periods of time (*p* < 0.05). The CA hydrogel was able to maintain the shape for 7 days in PBS after adding CaCl_2_ (data not shown).

### 2.6. Porosity of Collagen-Alginate Films

The porosity of collagen gel was significantly different (*p* < 0.05) from CA 4:1 and CA 7:3, with CA 4:1 containing the lowest percentage of porosity (Figure 4b). Other CA gels (CA 1:1, CA 3:7, and CA 1:4) were not significantly different (*p* > 0.05). The results were correlated with the SEM observations (Figure 2) and the swelling at 2 h and 4 h of incubation (Figure 4a). The CA porosity was proportional to the alginate concentration.

### 2.7. Rheological Properties of Collagen-Alginate Films

The rheological properties of collagen-alginate hydrogels in various ratio (CA 1:1, 1:3, and 3:1) was carried out in oscillatory and rotational mode. Frequency sweeps in all the tested gels behaved similar to a visco-elastic solid (Figure 5a), showing a storage modulus (G′) that was higher than lost modulus (G″) except CA 1:3. At 0.01 Hz, the storage modulus of CA 1:3, CA 1:1, and CA 3:1 was 172 Pa, 161.7 Pa, and 315.5 Pa, while the loss modulus was 33.1 Pa, 68.0 Pa, and 59.1 Pa, respectively. At 60 Hz, the storage modulus of CA 1:3, CA 1:1, and CA 3:1 was 351.5 Pa, 739.0 Pa, and 547.5 Pa, loss modulus was 122.5 Pa, 151 Pa, and 161 Pa, respectively.

In terms of amplitude sweeps, all the gels showed linearity between 0.01% and 20% strain with a slight decreasing trend thereafter except CA 3:1 (Figure 5b). The alginate did regulate the gel linear viscoelastic behavior.

To further enhance the rheology properties of the CA hydrogels, CaCl_2_ as cross-linker between concentration ranges of 0.4% (*w*/*v*) to 1% (*w*/*v*) were incorporated to the CA 1:1 gel, respectively. The viscosity in response to stress under various shear rates (0.1–100 s^−1^) were observed. The viscosity decreased with shear rate and exhibited a typical shear thinning behavior (Figure 5c). Generally, increasing the CaCl_2_ concentration helped enhance the zero-rate viscosity from 410.0 Pa⋅s to 775 Pa⋅s. Figure 5d shows the Tan δ for all the collagen-alginate hydrogels against frequency were below 1, except CA 1:3. The Tan δ increased with the alginate content in the gel mixture. A Tan δ value below 1 indicated the hydrogel behaves as a gel rather than a fluid-like mixture.

In the plot of shear stress against the shear rate (Figure 5e) the CA films show non-Newtonian behavior with intercepts at the shear stress axis. However, as the shear rate progresses, the Newtonian behavior was observed that resembles Bingham plastic. All hydrogels behaved as solids before stress was applied and turned to viscous liquid as the shear rate increased.

## 3. Discussion

Collagen from Ikan Belida was successfully extracted using an acid extraction method. However, the yield (0.25% dry weigh basis) that was obtained was rather low as compared to those that reported on fish skin collagen that was recovered from the same extraction method. Some reported fish skin collagen was about 10.9% (bigeye snapper) [41] to 54.3% (brown backed toadfish) [42], depending on the fish species. Low recovery from the current study might be due to improper handling in the fish ball processing plant, where the fish skin was not frozen immediately after the meat extraction. The protease degradation of the collagen resulted in low yield. Another reason might be the fish skin collagen was not fully solubilized in the acetic acid during the extraction procedure. An enzyme extraction method using pepsin may be another option to improve the collagen yield.

Fourier transform infrared (FTIR) was used to investigate the protein structure of collagen with the presence of alginate; the FTIR spectra is able to reveal the protein backbone of collagen. [43]. Amide I, II, and III signals on the CA films show the signature band for C-N stretching, N-H bending vibrations, and wagging vibrations of CH_2_ groups of glycine backbone and proline side chains in collagen [44,45]. Collagen is a protein build with repeating glycine, proline, and hydroxyproline arrangement. Therefore, the proline side chain that was detected on the CA films confirmed the existence of collagen. Amide I is the signature band for protein secondary structure, which was found in all the CA films [46]. A slight shift on the wavenumbers were observed with the increase of alginate in the collagen blend (Figure 1). The observation indicates interactions between the collagen and alginate. Alginate is an anionic polymer, which is in favor of establishing bonding between positively charged amide groups of collagens. This interaction is important in strengthening the intermolecular conformation and molecular structure of the CA film. This is believed to enhance the rheological properties of the collagen-alginate blend.

The cross-linkage between the collagen and alginate was further confirmed by the SEM images (Figure 2a–e). An egg box-like structure was observed as the alginate concentration increased in the CA (CA 1:1 and CA 3:7). The morphology differs from the pure collagen film, which shows a compact and rigid arrangement. The CA films were found to be porous as the alginate concentration increased. Similar morphological changes were observed by Zhou et al. [47] using sodium alginate-collagen hydrogels at the concentration ratio of sodium alginate: collagen from 1:4 to 4:1, respectively. These images indicated that the architecture of CA films can be altered by the combinations of collagen and alginate ratio to the desired pore size. The SEM images of collagen-alginate films are in agreement with the results of FTIR, swelling behavior, and porosity.

The denaturation temperature *(T_d_*) indicates the native conformation of the collagen in which the temperature that is higher than the *T_d_* results in the disruption of the collagen 3D structure. Collagen that is incorporated with alginate showed improvement in the denaturation temperature (*T_d_*). However, the alginate content beyond the optimum CA 1:1 has a lower denaturation temperature. Issains et al. [48] reported a similar trend where the melting point (*T_m_* ) increased with alginate in the collagen blend at a concentration between 1% and 3% with the optimum at 3% (110 °C). However, excessive alginate (5%) showed decreased in *T_m_*. The thermal stability of collagen is contributed by hydrogen bonding and facilitated by water molecules [49]. However, excessive alginate may weaken the interaction of some weak intermolecular forces such as the hydrogen bond between water molecules and the matrix. This, in turn, reduced the successful cross-linkage between collagen and alginate, resulting in the heat liable hydrogen bonds in the collagen structure being exposed to the surrounding. CA films from the present study showed *T_d_* that are higher than the physiological temperature (37 °C), which is especially important for the development of biomaterials in the future.

Another important property for the hydrogels is the swelling and porosity characteristics. These properties are important for cell proliferation in tissue culture. Changes in the microstructure of hydrogels may affect the migration of encapsulated cells and the ease of swelling allows the spreading of cells in the hydrogels around the wound. A high swelling hydrogel is also desirable due to its capabilities in preventing the breakdown of cell membranes (cell lysis) due to diffusion control of water molecules from the hydrogel-liquid interface [50]. The swelling properties correlate with the cross section of the SEM images (Figure 2), where the morphological structure varies as alginate content increased. Compared to the current observation with the 3D printed construct using an alginate collagen blend by Yang et al. [51], sodium alginate has the highest swelling properties followed by an alginate collagen blend and finally collagen agarose blend. Their printed constructs were able to maintain their structure after 48 h of incubation. However, most of the CA films in the current study were not able to achieve more than 24 h incubation, except CA 4:1 which retained its shape until 48 h. Alginate is a water-soluble polysaccharide, therefore, it was not surprising that CA films with higher alginate content dissolved in PBS after 24 h. However, the concentration of alginate (30 to 70 mg/mL) and collagen (10.5 to 24.5 mg/mL) that was used in the blend was rather low as compared to Yang et al. [51]. Increasing the alginate and collagen content to 0.1 g/mL and 15 mg/mL, respectively, as reported by Yang et al. [51] may have delayed the hydrogels from dissolving. Besides, CaCl_2_ was another factor that may alter the porosity of collagen-alginate gels. Moxon et al. [52] reported that the mean porosity of collagen-alginate gels was 25%, 20%, and 17% when 75 mM, 150 mM, and 300 mM CaCl_2_ were added, respectively.

In terms of the rheology study, increasing the collagen content generally increased the storage modulus at lower frequencies. However, the CA 1:1 gel overcame the storage modulus of CA 3:1 after approximately 30 Hz, this might be caused by insufficient cross-linkage that was established between the collagen and alginate. Therefore, this defect demonstrates weaker mechanical properties at a higher frequency.

Generally, increasing the CaCl_2_ concentration helped enhance the zero-rate viscosity from 410.0 Pa⋅s to 775 Pa⋅s. However, suppressed zero-rate viscosity was found in the CA blend with 1 % (*w*/*v*) CaCl_2_ that might be the due to the competition between the binding sites of alginate to CA^2+^ that disturbed the cross-linking establishment.

Rheological studies with the addition of CaCl_2_ to the hydrogels have shown solid structure at zero stress and transforms to a liquid state when stress was applied. This information, including the shear thinning behavior of the hydrogels, indicated possible important applications in the future. Shear thinning is an important characteristic in developing 3D printable hydrogels. An ideal 3D printable gels should possess a low viscosity when a shear force is applied, and the viscosity should recover immediately when the shear force is removed. Shear thinning behavior allows hydrogels to be fluid enough to flow through the printing needle during extrusion and withhold its structure after extrusion [53,54,55]. Yang et al. [51] reported that 3D printed prototype bioink that was prepared with 0.1 g/mL sodium chloride and 15 mg/mL collagen at the ratio of 4:1 yielded promising cell adhesion, enhanced cell proliferation, and enhanced cell expression of cartilage-specific genes (*Acan*, *Col2al*, and *Sox9*). Therefore, this hybrid collagen-alginate that was derived from marine waste products may help substitute the commercial pure collagen and alginate. However, further optimization of the collagen and alginate concentration is required to ensure that it is able to be utilized in human bodies.

## 4. Materials and Methods

The fish skin of Ikan Belida (*Notopterus lopis*) was obtained from a fish ball processing shop at Gotong Jaya, Pahang, Malaysia. Brown seaweed (*Sargassum polycystum*) was collected from Port Dickson Beach, Negeri Sembilan, Malaysia.

### 4.1. Extraction of Acid-Soluble Collagen (ASC) from Fish Skin

The method that was described by Nagai and Suzuki [56] was applied with minor modifications. All the preparation procedures were carried out at below 4 °C. The fish skin was first cut into smaller pieces of around (1 cm × 1 cm) to increase its surface area for collagen extraction. The fish skins were first deproteinized with 0.1 M NaOH at a sample/alkaline ratio of 1:8 (*w*/*v*) for 6 h. The solution was changed with a fresh solution every 3 h. The deproteinized fish skins were then repeatedly washed with cold distilled water until a neutral pH. Subsequently, the fish skins were defatted with 10 % butyl alcohol at a sample/solvent ratio of 1:10 (*w*/*v*) for 24 h, replacing with fresh solution every 12 h. The butyl alcohol residues were removed by rinsing the fish skin with cold distilled water until a neutral pH was obtained. The defatted skins were then soaked in 0.5 M acetic acid at a sample/solvent ratio of 1:2.5 (*w*/*v*) for 24 h. The acid-soluble collagens (ASC) were then recovered by filtering the skin residues with cotton cloth and the filtrate was recovered. The collagens were then salted out using 2.5 M NaCl in 0.05 M Tris buffer at pH 7.0. The white precipitates were then obtained by centrifugation at 14,000× *g* for 45 min at 4 °C. After that, the collected pellet was dissolved in a minimum volume of 0.5 M acetic acid and dialyzed against 50 volumes of 0.1 M acetic acid followed by distilled water for 24 h each.

### 4.2. Extraction of Alginate from Brown Seaweed

Alginate was extracted from brown seaweed (*Sargassum polycystum*) using the cold method that was described by Chee, Wong, and Wong [57]. Approximately 20 g of air-dried seaweed powder was soaked in 300 mL of 1% CaCl_2_ solution for 18 h at room temperature. Excessive CaCl_2_ was then removed from the seaweed by washing it with 300 mL distilled water 3 times. The seaweed powder was then subsequently treated in 5% HCl (300 mL) for 1 h. This treated seaweed was again rinsed 3 times with 300 mL distilled water to remove HCl residues. The rinsed seaweed was then soaked in 300 mL 3% Na_2_CO_3_ solution for 1 h at room temperature. A total of 250 mL distilled water was added then and the mixture was left overnight in room temperature. A viscous solution of soluble sodium alginate was obtained by centrifuging the mixture at 14,000× *g* for 10 min at 4 °C. The soluble sodium alginate was then precipitated with an equal volume of 95% absolute ethanol. The wet sodium alginate was filtered from its solution with cheese cloth and washed with 50 mL of 95% absolute ethanol. The alginate that was obtained was finally dried in a vacuum oven at 50 °C for 24 h.

### 4.3. Fabrication of Collagen-Alginate (CA) Hydrogels with Calcium Chloride (CaCl_2_)

Collagen-alginate hydrogels were fabricated according to the method by Sang et al. [58] with slight modifications. The extracted collagen was dissolved in 0.5 M acetic acid to yield 0.7% (*w*/*v*) monomeric solution by stirring for 2 h. The dissolved collagen was then neutralized to pH 7.2 with 2 M NaOH at 4 °C. Concurrently, alginate was dissolved in distilled water to create a 2% (*w*/*v*) solution. A collagen-alginate (CA) blend was obtained by adding the alginate solution drop wise to the neutralized collagen with continuous stirring for 2 h. The trapped bubbles were degassed for 10 min prior to being molded onto a Petri dish at 25 °C for 20 h to initiate gelation. The resulting hydrogels were washed several times with deionized water and stored in the freezer at −18 °C for at least 12 h. Finally, the frozen hydrogel was freeze-dried for 24 h to yield a collagen-alginate composite film.

The CA films were blended in the collagen: alginate ratio (*v*/*v*) as shown in Table 1 for further analysis.

The samples for rheological behavior were prepared with a CA hydrogel (1:1) that was blended with calcium chloride (CaCl_2_) solution at 0.4% (*w*/*v*), 0.5% (*w*/*v*), 0.75% (*w*/*v*), and 0.1% (*w*/*v*), respectively. Pure collagen and pure alginate film was fabricated, respectively, as experimental controls.

### 4.4. Fourier Transform Infrared Spectroscopy (FTIR)

Fourier transform infrared (FTIR) spectra of the collagen, alginate and CA films were obtained from an FTIR spectrometer (Thermo Scientific, Waltham, MA, USA). The spectra in the range of 400–4000 cm^−1^ with an automatic signal gain were recorded in 32 scans with resolution of 4 cm^−1^ at 25 °C and were done in ATR mode. Analysis was conducted in triplicates.

### 4.5. Scanning Electron Microscope (SEM)

The cross-sectional morphologies of all the CA films were observed using scanning electron microscope (SEM, Hitachi S-3400N, Chiyoda City, Tokyo, Japan). The films were coated in gold and scanned through the signal of SE (secondary electron) and BSE (back-scattered electron) at the working voltage of 10 kV with magnification of 200×.

### 4.6. Differential Scanning Calorimetry (DSC)

Differential scanning calorimetry (DSC) of collagen and CA films was performed using a differential scanning colorimeter (DSC 823e, Mettler Toledo, Nänikon, Switzerland) with slight modification from the method of Rochdi, Foucat, and Renou [59]. Indium standard was run to calibrate the temperature. Approximately 8 mg of the freeze-dried CA films were weighed into an aluminum pan and sealed. The collagen was scanned over the range of 20–50 °C, at scanning rate of 1 °C min^−1^. Whilst the CA films were scanned at 5 °C min^−1^ over the range of 25–180 °C [60]. An empty aluminum pan was used as a reference. The denaturation temperature (*T_d_*) of the collagen and CA films was determined from the first peak of the thermogram.

### 4.7. Swelling Behavior of CA Films

The swelling ratio of the CA film discs was obtained by firstly weighing the CA film discs (diameter 8 mm, thickness 1 mm) and are expressed as *W_dry_*. After that, the CA film discs were soaked in a sealed tube filled with 5 mL 1 × PBS at pH 7.4 and 37 °C [58]. The samples were removed and blotted with filter paper at an interval of 2, 4, 6, 24, and 48 h, respectively. The swollen samples were weighed as *W_wet_*. The swelling ratio (*S*) of each CA film disc was estimated by using the equation below:(1)$$Swelling\;\left(\%\right)=\frac{W_{wet}-W_{dry}}{W_{dry}}\times 100$$

### 4.8. Porosity of CA Films

The porosity (*D*) of all the CA films were measured according to the method that was used by Sang et al. [58]. The geometrical volume (*V_s_*) of the CA discs was estimated by measuring the height and diameter, whilst the pore volume (*V_p_*) was determined through ethanol displacement method. The CA discs were weighed (*W_o_*) prior to soaking the discs in absolute ethanol. Subsequently, the discs were placed in a desiccator under vacuum for 5 min. The samples were then removed from the desiccator and blotted gently with filter paper. The weight (*W_e_*) was recorded immediately. The porosity of the CA films was calculated as:(2)$$D=\frac{V_{p}}{V_{s}}\times 100$$
where, $V_{p}=\frac{W_{e}-W_{o}}{\rho_{e}}$, $\rho_{e}$ represents the density of ethanol (0.789 mg mL^−1^).

### 4.9. Rheology Properties of Collagen-Alginate Hydrogels

Rheological testing of collagen-alginate films (CA 1:3, CA 1:1, and CA 3:1) was done based on the method that was applied by Duan et al. [61]. An Anton Paar Physica MCR 301 Rheometer (Graz, Austria) was used for the rheological testing. An amplitude sweep was done to obtain its linear viscoelastic range (LVE). The strain was set from 0.01% to 100% at a constant frequency of 1 rad/s. A frequency sweep was then done to determine the hydrogel behavior. The hydrogel was then subjected to a frequency range of 0.01 Hz to 10 Hz with an amplitude strain within the LVE. All the tests were done at 25 °C with a 25 mm flat parallel plate measuring pole and a 0.5 mm gap. Solvent traps were not used in the current study due to a short testing period. The samples were analyzed in triplicate.

### 4.10. Statistical Analysis

Analysis was carried out in triplicate and the results were reported as the mean values with standard deviation except for FTIR, SEM, and rheology properties. Different mean values were analyzed by analysis of variance (ANOVA). A difference between means at the 5% levels was considered as significant.

## 5. Conclusions

Hydrogels from Ikan Belida fish skin collagen and brown seaweed alginate were characterized in various physical and rheological aspects. The alginate was able to alter the physical characteristics of collagen to a porous and flaky structure as seen in the SEM. The *T_d_* was improved to 83.7 °C (CA 7:3), however, excessive alginate (CA 3:7) did not exhibit high *T_d_* as expected. The swelling and porosity of CA were correlated with the SEM images. Generally, the alginate content affected the swelling and porosity as shown in the SEM, where the alginate content resulted in porous gels with high water absorption ability. However, due to high alginate content, the gels dissolved in 2 h. CA 1:1 demonstrated better absorption ability, maintaining its structure stability within 48 h. The hybrid hydrogels exhibited visco-elastic solid and shear thinning behavior. The results from the rheology properties study serve as the groundwork for the development of biomaterial or hydrogels for bioprinting. The viscosity and the viscoelasticity of the hydrogels can be manipulated by the formulation between collagen, alginate, and CaCl_2_. The study confirmed that collagen from fish skin of Ikan Belida and alginate of the local brown seaweed has the potential to be incorporated into the downstream processing route of the local marine product industries. A collagen-alginate hydrogel that was developed in this experiment was shown to have good potential to be applied in bio fabrication.

## Figures and Tables

**Figure 1 marinedrugs-20-00525-f001:**
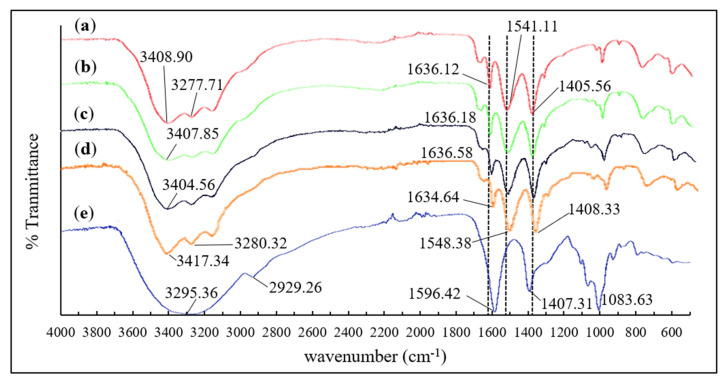
FTIR spectra of (**a**) Pure collagen film; (**b**) CA 7:3; (**c**) CA 1:1; (**d**) CA 3:7; (**e**) Pure alginate film.

**Figure 2 marinedrugs-20-00525-f002:**
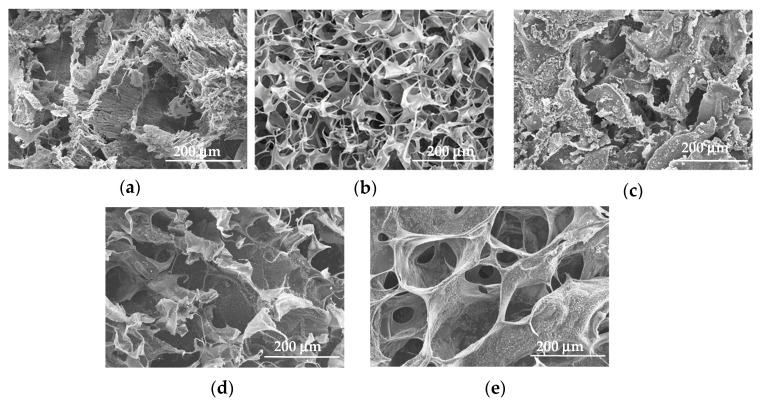
SEM collagen-alginate films at various ratios. (**a**) Pure collagen film; (**b**) Pure alginate film; (**c**) CA 7:3; (**d**) CA 1:1; (**e**) CA 3:7.

**Figure 3 marinedrugs-20-00525-f003:**
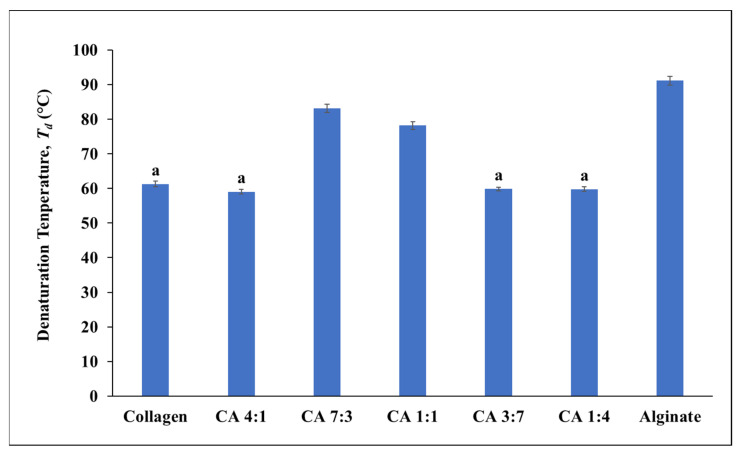
Denaturation temperature (*T_d_*) of collagen-alginate film in various ratio. The error bars represent the standard deviation of the mean. Means sharing the same letters are not significantly different from each other (*p* > 0.05) as indicated by Tukey’s test.

**Figure 4 marinedrugs-20-00525-f004:**
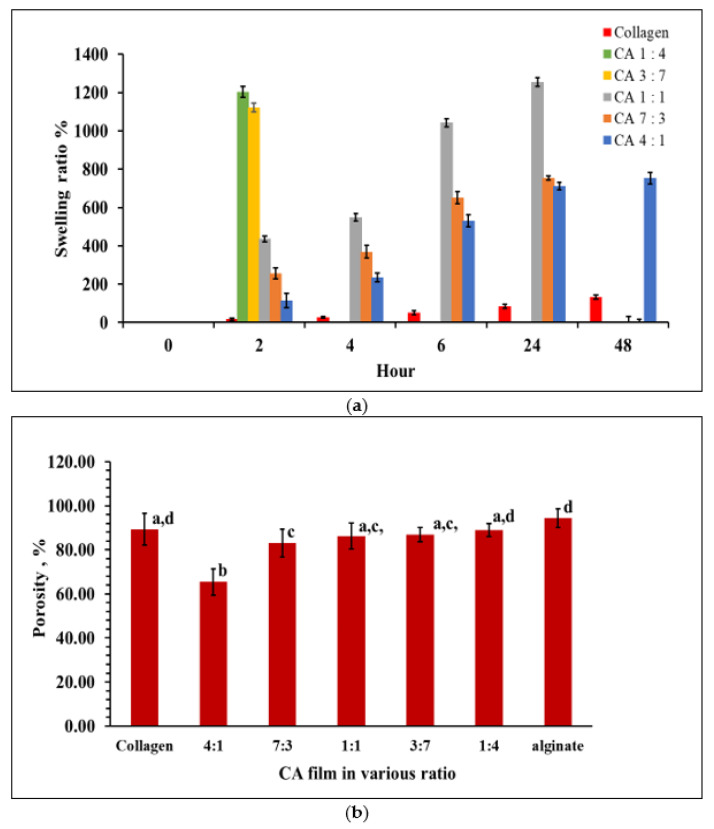
(**a**) Swelling ratio of collagen-alginate gels; (**b**) comparison of the porosity and swelling of collagen-alginate films. The error bars represent the standard deviation of the mean. Means sharing the same letters differ are not significantly different (*p* > 0.05) as indicated by Tukey’s test.

**Figure 5 marinedrugs-20-00525-f005:**
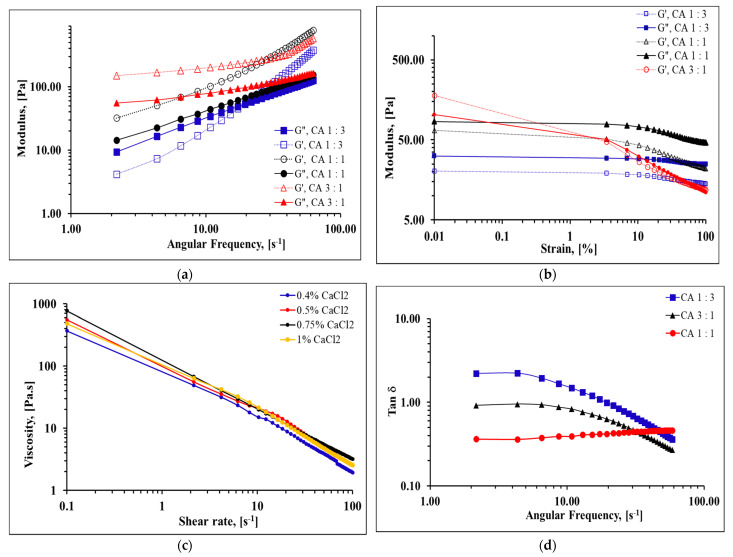
Rheology characterization of collagen-alginate hydrogels in various ratios. (**a**) The storage modulus G′ and loss modulus G″ under different angular frequencies; (**b**) The storage modulus G′ and loss modulus G″ at different oscillation time sweep; (**c**) Viscosity of collagen-alginate hydrogels (CA 1:1) with different amounts of CaCl_2_ at various shear rates; (**d**) Tan δ at various frequency; (**e**) Shear stress of the collagen-alginate gels in response to temperature.

**Table 1 marinedrugs-20-00525-t001:** Composition of the collagen-alginate mixtures.

Designation	V_collagen_ (vol %)	V_alginate_ (vol %)
CA 4:1	80	20
CA 7:3	70	30
CA 1:1	50	50
CA 3:7	30	70
CA 1:4	20	80

## Data Availability

Not applicable.

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
