# Peer review of "Utilizing Fish Skin of Ikan Belida (Notopterus lopis) as a Source of Collagen: Production and Rheology Properties"

_marinedrugs, 2022, doi:10.3390/md20080525_

Round 1

Reviewer 1 Report

The work herein reported is interesting from the concept point of view, although the combination of collagen with alginate to generate hydrogels is not new, considering the fish species from which the collagen is extracted. Moreover, several issues must be addressed to improve the quality of the work, some of which are associated with important limitations in the study and/or the obtained results, as follows.

1 - In abstract (line 21), it is stated that the SEM analysis revealed porous and flaky structures. Wouldn't this be mostly a result from freeze-drying of the samples?

2 - Line 46: The similarity with type I collagen is not of the sources, but of the collagen isolated from those sources. It should be rewritten as something like "These sources enable the production of proteins that are often similar to type I collagen..."

3 - The advantages of fish collagen associated with circular economy should be also mentioned. Not being unique from fish (it's also the case when using mammals hides, for instance), it is an important issue.

4 - Line 61: The term "bioinks" should be restricted to printable materials laden with cells. If only the materials are present, than inks or biomaterial inks should be used instead.

5 - Line 75: Is it "printing needle" or "printing nozzle"?

6 - Lines 75/76: Maybe the authors could relate the need to maintain shape after printing with the requiring of shear thinning behavior and/or post-processing to assure that stability.

7 - Lines 108 and 109: Please identify to which "former" and "latter" are referring to.

8 - Line 110: This sentence seems to be repeated.

9 - Line 112: The statement that RGD motifs help in wound healing needs further explanation. RGD is associated with cell adhesion and this may help on wound healing, but I don't think a direct relation can be assumed...

10 - Line 137: The extraction yield of 0.25% seems to be quite low. How does it compare with other species?

11 - The shifts highlighted in FTIR peaks seem to be quite low to be really significant, as they are - in some cases - smaller that the resolution of the spectra.

12 - Line 173: A denaturation temperature of 61.26 ºC to fish collagen seems very high, considering other values being reported in literature for other fish collagens (around 15 to 30 ºC)... Maybe this is due to the method used for its determination. Besides, why the formulations CA 3:7 and CA 1:4 did not show an increase in Td? Might it be an indication of not effective crosslinking? Please comment.

13 - Why different collagen:alginate ratios were used for rheological analysis when compared with the other analyses? Why not keeping the 4:1, 7:3, 3:7 and 1:4?

14 - Why in discussion there was no comments on the extraction and characterization of collagen? Wasn't it the first time this fish species was used for collagen isolation?

15 - Lines 256 to 261: This paragraph is quite confusing. Please revise it. What does it means the increase of pore distribution? Wider range of pore sizes? Uniform porosity? Other?

16 - Lines 280/281: If the hydrogel does not withstand 24h incubated, it is an indication of not appropriate/efficient cross-linking. Thus, the produced hydrogels are not appropriate for cell culture, the proposed application, does questioning the rational of the study. This is a major flaw in the study and must be properly addressed by the authors.

17 - Lines 287/288: To which effect is this sentence referring to? Besides, the relation between porosity and concentration of calcium chloride would certainly depend also on polymer concentration, among others and thus care should be taken when comparing different studies.

18 - The rheological results seem to indicate that some formulations did not even support continuous (and "vigorous") manipulation of the hydrogel without destroying it. Indeed, the authors indicate in lines 312/313 that the concentration of collagen and alginate needs to be improved, from which I believe that the improved or optimized formulation is required to give proper significance to the study.

19 - Regarding the preparation of the hydrogels, was there any physical change in the collagen solution when neutralizing it to pH 7.2? How did the gelation occur later on? Wasn't CaCl2 needed to jelify alginate? Or was it "simply" by electrostatic interaction between charged groups in collagen (+) and alginate (-)?

20 - Why both secondary electrons and back-scattered electrons signals were used and not simply the secondary electrons signal, as the samples were coated with gold to be conductive?

21 - Regarding the DSC analysis, could you please clarify if the CA samples were in dry state or hydrated?

22 - How the analysis of the swelling behavior took into consideration the lack of structural stability in aqueous media?

Reviewer 2 Report

The authors have studied a fish-derived collagen.  The collagens from perhaps more than a hundred different fish have now been described.  It is important therefore that studies are extended to give an extra level of information.  The authors have achieved this, through their examination of collagen alginate composites and by their focus on a circular economy, where waste is given value.  This could be stressed more?

My main concern with the present manuscript is that there are places where the English style is needing revision.  In its present format it is distracting to read in places.  

Otherwise, the authors have collected a significant amount of data and have described these results.  However, I find that the extent of the Discussion on the relevance of the data is very limited in places.  For example, the relevance of the FTIR needs extending.  The pore sizes observed, such as the SEM, could be described further and compared to those needed for different applications, etc.

The Abstract starts with ‘mechanical strength’ but seems that mechanical properties (eg: Youngs Modulus, etc) are not examined, just rheological properties?  Later, in the Introduction, this theme is revisited, toughness and durability, prior to a discussion on rheology.  But these aspects are not covered in the experimental focus, and their inclusion should be reconsidered.

There are a few specific points.  The authors described the material as “biocompatible”.  Biocompatibility is not a generic property of a material, but rather reflects the response in a specific application.  (See D Williams’ Dictionary and similar texts.)   Thus, your collagen, if used for a vascular coating would lead to thrombosis, and would not be considered biocompatible!  So, the various sentences using this term need modifying so that a generic, incorrect statement is avoided.

Around line 122, the economics are discussed.  This is in a local context.  More generically it is true “for certain economies” and the text could be modified.

Line 137.  Mentions the “acid hydrolysis method”, where a better description would be acid extraction as it is solubilisation that is happening, not a hydrolysis step.

Line 270.   I am not sure why a low Td is preferable?   Rather, the Td should exceed human physiological temperature, so as to avoid denaturation.  And, typically for fish derived collagens, their melting temperatures are lower than human body temperature, reflecting the marine environmental temperature.  (See B Rigby papers for example).  Stabilisation is needed, to raise the Td.

Also, the option of bioprinting is often included in the text, especially in the introduction, but the data do not describe any printing experiments, but rather discuss films/sheets of material.  The text needs reviewing.  The Introduction should be more concise, such as if it were to focus on the experimental directions.

Reviewer 3 Report

The paper studies the preparation of biomaterials from Collagen of fish and alginate from algae The paper shows well the characterization of the material with the physical and optical techniques. The techniques are adequate and well used.  It is known the low mechanical resitance of the collagen that is improved by the addition of alginate, to confer better strucutral properties as is shown in the paper.

Some concerns:

It is not well justified why it is so important the rheology of the mixes. I work with rheology and sometimes is difficult to explain the people why this techniques are important for the final material. In this paper, the rheological analysis is adequate, but it needs more explanation to relate the rheological results to the characteristics of the hidrogel.

It would be interesting also to see some application. It is claim the use in biomedicine, but what kind of use? It is true that is difficult to develop one application , but al least qualitatively made some indications about the improvement reached for the introduction of the alginate in the collagen materials.

Figure 3- It is extrange, that when the quantity of alginate is higher the denaturation temperature is closer to the collagen temperature, it seems not logic. Some suggestion?

Figure 4.a, what happen with CA 1:4 after two hours? The hydrogels colapse?. And the CA1:1 after 24 hours. It seems that the estability of the materials is not high. If the time in the X-axis is another time, it must be explained, because is difficult to understand.

Lines 301-313 the indicaton of the biomaterial as possible ink is adequate but, as is mentioned in the last sentence it need optimization, not only to determine the technical utilization in the needle of the 3D printers but also in the type of product printed and their use.

Round 2

Reviewer 3 Report

The manuscript have been corrected accordingly, following the indications of the reviewers